# Multi-Functional Electrospun AgNO_3_/PVB and Its Ag NP/PVB Nanofiber Membrane

**DOI:** 10.3390/molecules28166157

**Published:** 2023-08-21

**Authors:** Taohai Yan, Shengbin Cao, Yajing Shi, Luming Huang, Yang Ou, R. Hugh Gong

**Affiliations:** 1Fujian Key Laboratory of Novel Functional Textile Fibers and Materials, Minjiang University, Fuzhou 350108, China; thyan@mju.edu.cn (T.Y.); SYj728163@163.com (Y.S.); yangou2000@hotmial.com (Y.O.); 2School of Materials, Shanghai Dianji University, Shanghai 201306, China; 3Department of Materials, University of Manchester, Manchester M13 9PL, UK; luminghuang@hotmial.com (L.H.); hugh.gong@manchester.ac.uk (R.H.G.)

**Keywords:** electrospinning, PVB fiber, AgNO_3_, Ag NPs, anti-UV, anti-electromagnetic radiation

## Abstract

This study focuses on the fabrication of fiber membranes containing different concentrations of AgNO_3_ via the electrospinning technique. The AgNO_3_ present in the fibers is subsequently reduced to silver nanoparticles (Ag NPs) through UV irradiation. The resulting nanofiber film is characterized using scanning electron microscopy, X-ray diffraction, and evaluations of its anti-UV and anti-electromagnetic radiation properties. Experimental results demonstrate that increasing the AgNO_3_ content initially decreases and then increases the fiber diameter and fiber diameter deviation. Under UV light, the nanofibers fuse and bond, leading to an increase in the fiber diameter. AgNO_3_ is effectively reduced to Ag NPs after UV irradiation for more than 60 min, as confirmed by the characteristic diffraction peaks of Ag NPs in the XRD spectrum of the irradiated AgNO_3_/PVB fibers. The nanofiber film containing AgNO_3_ exhibits superior anti-UV performance compared to the film containing AgNO_3_-derived Ag NPs. The anti-electromagnetic radiation performances of the nanofiber films containing AgNO_3_ and AgNO_3_-derived Ag NPs are similar, but the nanofiber film containing AgNO_3_-derived Ag NPs exhibits higher performance at approximately 2.5 GHZ frequency. Additionally, at an AgNO_3_ concentration of less than 0.5 wt%, the anti-electromagnetic radiation performance is poor, and the shielding effect of the nanofiber film on medium- and low-frequency electromagnetic waves surpasses that on high-frequency waves. This study provides guidance for the preparation of polyvinyl butyral nanofibers, Ag NPs, and functional materials with anti-ultraviolet and anti-electromagnetic radiation properties.

## 1. Introduction

Affected by a new heatwave, many countries in the northern hemisphere, including Italy and Spain, continue to experience high temperatures. Scientists suggest that 2023 could be the hottest year on record. Textiles carry the consumers’ aspirations for a better life, with the hope that they possess multifunctional properties, such as ultraviolet (UV) resistance and antibacterial features. Silver nitrates and their conversion to nanosilver have many excellent properties, making them one of the most investigated research materials. Given that absorbing UV light converts silver nitrate into nanosilver, its addition into fibers will enhance the UV shielding performance of existing materials. Polyvinyl butyral (PVB) is a thermoplastic encapsulant widely used in various applications, including safety glass laminates, building-integrated photovoltaics, and thin films with a glass-to-glass configuration [1,2]. PVB is frequently employed in electrospinning due to its adjustable molecular weight and excellent spinnability [3,4,5]. Electrospun PVB nanofibers have found applications in bioengineering fields, serving as antimicrobial agents, drug carriers, and tissue engineering materials, owing to their antimicrobial properties, biocompatibility, and non-cytotoxicity [6]. Furthermore, PVB fibers are used in the production of oil, thymol, berberine hydrochloride (BH) carriers for atopic skin treatment, as well as in face masks and air filtration [7]. PVB fibers also serve as composite matrix materials due to their stability and ease of removal, enabling the preparation of various composite materials, such as mesoporous alumina nanofibers and pure γ-Y_2_Si_2_O_7_, through electrospinning [8,9,10]. Additionally, PVB has been utilized as a substrate for liquid-assisted quasi-solid polymer electrolytes, organic light-emitting devices, and magnetic Fe_2_O_3_ nanoparticles, showcasing its stability and performance [11,12].

Electrospun nanofiber membranes incorporating AgNO_3_ have demonstrated excellent antibacterial activity and electrical conductivity when combined with polymers such as PAN, PVDF, and polycaprolactone. Among them, polyacrylonitrile (PAN) is a common polymer fiber in laboratory and industrial production. It has excellent thermal stability and chemical resistance, making the electrospinning process easier to control when used as a polymer in electrospinning. These nanofibers have been applied in fields such as air and wastewater filtration, tissue engineering, and biomedical engineering [13,14,15,16]. Many methods were studied for the conversion of silver nitrate to nanosilver. In recent years, the preparation of silver nanoparticles (Ag NPs) has advanced rapidly. Ag NPs possess unique electrical, optical, and catalytic properties, making them suitable for applications such as catalysts, new electrode materials for batteries, electrically conductive materials, and low-temperature thermally conductive materials [17,18,19,20]. Ag NPs can be prepared through physical or chemical methods. Physical methods include sputtering, vacuum evaporation, and laser ablation, which are relatively simple mechanisms but require high operating conditions and are expensive to perform. Gao et al. prepared the electrospun Panax notoginseng@Ag core–shell fiber membrane by coaxial electrospinning combined with the UV reduction method (254 nm). The resulting silver nanoparticles had antibacterial effects, synergistically promoting rapid wound healing [21]. Using UV light was a safe and interesting method of conversion in this paper. Chemical methods, on the other hand, include microemulsion, chemical reduction, electrochemical reduction, ultrasonic reduction, and photochemical reduction, offering advantages such as ease of control, simple procedures, and the ability to produce Ag NPs with small sizes [22,23,24,25]. Ag-NP-loaded nanofibers, particularly those prepared using PAN as the polymer, have exhibited high antimicrobial activity and great potential for biomedical and pharmaceutical applications. In a study by Tang et al., nanofibrous mats were treated with silver nitrate. Silver nitrate was chemically reduced to silver nanoparticles, and AgNP-HHK/PEO/PVA nanofibrous mats exhibited excellent antibacterial activity [26]. The size, distribution, and loading density of Ag NPs on other polymer nanofibers can be easily controlled by adjusting the AgNO_3_ concentration, which significantly influences the antibacterial properties of composite nanofibers. Moreover, AgNO_3_ and Ag NPs can serve as conductive materials for dynamic sensor applications and Raman spectroscopy substrates [27]. Nanosilver possesses many excellent properties, enabling multifunctionality within fibers.

PVB, as a material possessing anti-ultraviolet properties, exhibits a stable structure suitable for use as a matrix material [28,29]. Very few researchers have utilized AgNO_3_/PVB, and the main research focus has been on its antibacterial properties. Yalcinkaya et al. prepared AgNO_3_/polyvinyl butyral (PVB) electrospun fibers, where silver nanoparticles were prepared by reducing silver salt using ascorbic acid. Results showed that higher PVB polymer solution concentrations had better efficiency against Escherichia coli (*E. coli*) [30]. More studies reported that the incorporation of nanosilver or silver ion compounds into PVB materials improves their antibacterial, electrical conductivity, and anti-ultraviolet properties, expanding their potential applications. For instance, Barrera et al. developed an Ag-NP/PVB polymeric nanocomposite printable fluid using silver nanoparticles and polyvinyl butyral (Ag-NP/PVB), which demonstrated bacterial inhibition [31]. This novel composite is an exciting material with functionalities such as UV protection, electromagnetic shielding performance, potential silver antimicrobial properties, etc. that can be applied to functional textiles. As mentioned earlier, PVB is a suitable polymer for electrospinning, allowing for the production of nanofibrous Ag-NP/PVB materials with permeability, breathability, structural integrity, and increased potential applications in biomedicine, sensors, filtration, and ultraviolet and electromagnetic radiation protection. The summer of 2023 has been the hottest on record in many places. Many new cars, such as Teslas, now have beautiful panoramic glass roofs. However, they have high UV transmission rates and do not provide adequate heat insulation, resulting in high temperatures inside the vehicles. The intense sunlight can make occupants feel uncomfortable. The findings of this study may provide valuable insights into improving the UV resistance of double-layer PVB glass.

Current academic research mainly focuses on the antibacterial properties of silver nitrate and nanosilver, with relatively fewer studies on their anti-ultraviolet functionalities. PVB itself has certain UV-resistance properties. As extreme weather becomes more frequent, the demand for UV resistance in textiles and glass products is increasing. Therefore, this paper introduces silver nitrate into PVB to enhance its multi-functional properties. We prepared fiber membranes with different concentrations of AgNO_3_ via electrospinning and subsequently reduced the AgNO_3_ to Ag NPs through UV irradiation. We investigated the anti-ultraviolet and electromagnetic radiation protection performances of nanofibers with silver nitrate and nanosilver incorporated into PVB.

## 2. Results and Discussion

### 2.1. Morphological Analysis

Figure 1 displays the SEM images of the nanofiber membranes prepared with different concentrations of AgNO_3_ and subsequently UV-irradiated for 60 min. Figure 1(1b,2b,3b,4b,5b) shows the SEM images of the non-irradiated nanofiber membranes with corresponding AgNO_3_ concentrations. The fiber diameters for the 0%, 0.5%, 1%, 1.5%, and 2.0% AgNO_3_ samples were measured to be 0.26 ± 0.03; 0.22 ± 0.02; 0.20 ± 0.01; 0.19 ± 0.01; and 0.24 ± 0.02 µm, respectively. From Figure 2a, it can be seen that the concentration of AgNO_3_ is less than 2.0% and that the fiber diameter does not change much. Fibers without AgNO_3_ exhibited slightly greater thickness and occasional beading. Conversely, the electrospun fibers containing AgNO_3_ appeared smooth, straight, and devoid of beading, with a highly uniform fiber diameter distribution. With an increasing AgNO_3_ content, the fiber diameter and deviation initially decreased and then increased. The improved conductivity of the spinning solution due to AgNO_3_ led to greater stretching of the jet under a stronger electric field force, resulting in smaller fiber diameters. However, this trend was not infinite, as there is a limit to how much repulsive charges can affect the viscoelastic force of the solution. Beyond this point, further increases in salt content did not lead to additional reductions in fiber diameter. Previous studies have reported an initial decrease in fiber diameter with the addition of AgNO_3_, followed by an increase as AgNO_3_ content further increased in aqueous-based polymer solutions. Moreover, the quality of the fibers deteriorated with increased beading and variation in fiber diameter at higher AgNO_3_ concentrations [32]. As Ke H et al. reported, the diameters of the obtained PAN/AgNP composite fibers decreased with an increase in the initial AgNO_3_ concentration in the solutions [33].

After UV irradiation, the fiber diameter decreased, and irregularities in diameter increased due to the reduction in material mass after decomposition. Under UV light, the PVB fibers experienced heating and fusion, resulting in an increase in fiber diameter, as shown in Figure 1. This fusion was attributed to the glass transition temperature of PVB, which ranges from 50 °C to 70 °C [34]. The UV-irradiated fibers without AgNO_3_ exhibited the highest fusion degree, as depicted in Figure 1a, while the UV-irradiated fibers with 2.0 wt% AgNO_3_ exhibited the lowest fusion degree. This study used a UV aging machine as a standard light source for UV irradiation, and all samples were stored in a specific enclosed space. After UV irradiation, the sample heating phenomenon occurred; hence, the fibers in the samples melted and stuck together, which is a distinctive event. In contrast, other papers used relatively lower UV irradiation intensities and different glass transition temperatures. In addition, their samples were placed in a more open space, where heat could disperse easily and not heat the fibers to their melting point, thus avoiding the melting and sticking phenomenon [33,35,36]. This indicates that the fusion degree decreased with increasing AgNO_3_ concentration. 

The AgNO_3_ absorbed energy from the UV light and underwent a chemical reaction (Equation (1)), reducing the overall temperature of the nanofiber membrane and decreasing the number of fused PVB fibers [37]. The average fiber diameter after irradiation was larger than before irradiation due to fiber adhesion and irregularities caused by fusion. Additionally, AgNO_3_ in the PVB fibers absorbed UV light and converted to Ag NPs, contributing to a reduction in the overall temperature of the nanofiber membrane.
(1)$${2AgNO}_{3}\overset{UV}{\to }2Ag+{2NO}_{2}↑+O_{2}↑$$

From Figure 2b, it can be seen that with the increase in radiation time, the diameter of the nanofiber membrane containing 1.5 wt% AgNO_3_ shows an upward trend. Figure 3 present SEM images of the nanofiber membrane containing 1.5 wt% AgNO_3_ and subjected to different UV irradiation times: 0, 60, 120, 180, and 240 min. As the UV irradiation time increased, a greater amount of AgNO_3_ underwent the reaction, leading to the formation of Ag NPs. Consequently, the temperature of the irradiated fibers naturally increased. At the glass transition temperature, a significant number of PVB fibers fused and adhered to each other, resulting in the observed structural changes. In addition, it was found that the surface color of the fiber membrane gradually changed from light yellow to dark brown. Similar to the report by Hong K H et al., all the colors of the electrospun AgNO_3_-containing PVA composite fiber webs changed from a white tone to a light yellow tone during the preparation process. This indicates that in the electrospun AgNO_3_/PVA fibers, Agþ ions were reduced and aggregated into Ag nanoparticles [35].

### 2.2. XRD Analysis

The X-ray diffraction (XRD) spectrum of the non-irradiated AgNO_3_/PVB fibers did not exhibit any diffraction peaks, as shown in Figure 4, indicating the absence of a crystal structure or Ag NPs within the fibers. Comparatively, the XRD patterns of the fibers irradiated for 60, 120, and 180 min displayed characteristic diffraction peaks associated with Ag NPs. The XRD patterns confirmed that the Ag NPs within the Ag/PVB fibers possessed a face-centered cubic structure. Notably, the XRD spectrum of the irradiated AgNO_3_/PVB fibers displayed distinct peaks at 2θ values of 38.04°, 44.28°, 64.46°, and 77.42°, corresponding to the 111, 200, 220, and 311 crystallographic planes of face-centered cubic Ag crystals, respectively. The distinct peaks were consistent with those shown in the paper of Hassanien A S et al. [38]. These results indicated that the Ag NPs formed were crystalline. Conversely, the XRD patterns of the non-irradiated AgNO_3_ samples revealed the presence of Ag as the main crystalline phase, without any additional phases or impurities (Ag XRD Ref. No. 01-087-0719) [39]. It can be concluded that AgNO_3_ was successfully transformed into Ag NPs after UV irradiation for more than 60 min. Pure nanosilver particles had no impurity X-ray diffraction (XRD) peaks [40]. The nanosilver particles in this study were encapsulated in PVB polymer, and similar to the report of Hong K H et al. on AgNO_3_/PVA, after crystallization, the impurity peaks (at around 2θ = 23°) of PVB could be observed [35]. The crystallinity of the PVB polymer was increased after UV treatment. This is in contrast to the paper of Hong K H et al., where the characteristic nanosilver peaks were not clearly visible. The reason was that the temperature they used for treating silver nitrate was only 155 °C, which had not reached the decomposition temperature of silver nitrate [41]. However, in this study, the silver nitrate decomposition was more thorough due to the relatively high UV light intensity, and the nanosilver characteristic peaks were clearly visible.

The average particle size of the Ag NPs was estimated using the Debye–Scherrer equation (Equation (2)) [42]:(2)D=Kλ/(βcos θ)
where *D* represents the average crystallite size of the Ag NPs, *K* is the Scherrer constant (with values ranging from 0.9 to 1, accounting for the shape factor), *λ* is the X-ray wavelength (1.5418 Å), *β*1/2 is the width of the XRD peak at half height, and *θ* is the Bragg angle. Based on the Scherrer equation, the average crystallite size of the Ag NPs was estimated to be approximately 10–20 nm, and the calculated average crystallite size aligned with the grain size range computed by other researchers [43,44].

### 2.3. Anti-UV Performance Test and Analysis

The ultraviolet protection factor (UPF) and ultraviolet transmittance of the nanofiber films were measured using an anti-UV performance tester according to the AS/NZS 4399:1996 test standard [45]. The anti-UV coefficient, which represents the ratio of the average amount of UV radiation absorbed on the unprotected skin to the amount absorbed on the tested textiles, was used to evaluate the effectiveness of the UV-irradiated textiles.

Due to the inherent anti-UV function of PVB fibers, the UPF value of the PVB fibers containing AgNO_3_ was already above 40, indicating excellent UV protection capability, as exhibited in Table 1, which reveals an enhancement in the anti-UV function of PVB materials with the addition of AgNO_3_ or Ag NPs. The nanofiber film containing AgNO_3_ at concentrations ranging from 0.5% to 1.5% exhibited better anti-ultraviolet performance than the nanofiber film containing Ag NPs. This result can be attributed to AgNO_3_ absorbing UV light and converting it into Ag NPs, thereby reducing the amount of UV radiation penetrating the nanofiber film. This effect was most pronounced at an AgNO_3_ concentration of 0.5%. The ultraviolet transmittance of the non-irradiated nanofiber membrane was significantly lower than that of the nanofiber membrane containing AgNO_3_-derived Ag NPs. Similarly, the UPF value of the non-irradiated nanofiber membrane was significantly higher than that of the membrane containing AgNO_3_-derived Ag NPs. This difference can be attributed to the smaller content of AgNO_3_-derived Ag NPs, which was insufficient to cover the entire nanofiber membrane [46]. At an AgNO_3_ concentration of 2%, the anti-UV performances of the irradiated nanofiber films containing AgNO_3_ and AgNO_3_-derived Ag NPs were similar. The nanofiber film containing AgNO_3_-derived Ag NPs exhibited a high UPF value, indicating the effective blocking of UV rays through reflection, scattering, and light absorption. In theory, the higher the AgNO_3_ concentration, the greater the anti-UV performance of the nanofiber membrane. Notably, UV protection factor (UPF) values exceeding 100 were displayed as 100+ due to the limitations of the testing instrument. The actual UPF values of the irradiated nanofiber films with higher AgNO_3_ concentrations were much higher than 100.

In existing studies on the UV resistance of Ag NP textiles, Ag NPs were attached to the fabric surface through a post-processing technique. For example, after loading nanosilver onto cotton textiles with poor UV protection, the UV protection factor (UPF) value was 148, and the transmittance rate for UVA was 1.11% [47]. Also, after depositing nanosilver onto the surface of leather textiles, which inherently had good UV protection, the UPF value (478 ± 3) was significantly higher than other samples [48]. However, these post-processing techniques did not typically have lasting functionality, and the functionality significantly decreased after washing [49]. The nanosilver in this study was added to the fibers during the electrospinning process, and silver nitrate itself could absorb UV rays. After conversion to nanosilver, its UV resistance was outstanding. Due to the range limitations of equipment, the UPF value was 100+, but the transmittance rate for UVA was just 0.03, and its UV protection far exceeded other samples. Additionally, this functionality was permanent and did not decrease after washings. This excellent UV-resistant functionality makes it particularly suitable for scenarios where high UV protection levels are required.

### 2.4. Electromagnetic Radiation Protection Performance Test and Analysis

The electromagnetic radiation protection performance of the nanofiber films was evaluated using a DR-913G fabric anti-electromagnetic radiation performance tester [50], and the data obtained were plotted on a broken line chart. Figure 5 illustrates the slight differences in the anti-electromagnetic radiation performance between the non-irradiated and irradiated nanofiber films, indicating slight variations in conductivity between the nanofiber films containing AgNO_3_ and AgNO_3_-derived Ag NPs. However, the nanofiber film containing AgNO_3_-derived Ag NPs exhibited superior anti-electromagnetic radiation performance compared to the non-irradiated nanofiber film containing AgNO_3_, particularly around the frequency of 2.5 GHZ [51,52]. This finding suggests that the nanofiber film can be utilized for targeted electromagnetic radiation protection at specific frequencies. The comparison of transmittance among the fiber membranes with different AgNO_3_ concentrations revealed poor anti-electromagnetic radiation performance at AgNO_3_ concentrations below 0.5 wt%, indicating inadequate conductivity in these membranes. Conversely, AgNO_3_ concentrations above 1 wt% yielded good anti-electromagnetic radiation performance, although the differences were not significant. Overall, the nanofiber membrane exhibited better shielding effects against medium and low-frequency electromagnetic waves compared to high-frequency waves [53]. Due to their simplicity and efficient results, the preferred techniques for incorporating silver nanoparticles were direct blending and ultraviolet irradiation methods [36]. 

Current research on the electromagnetic shielding effect of metal nanoparticles on textiles primarily focuses on the loading of metal nanoparticles on the surface of the textiles; the ability to shield electromagnetic (EM) waves depends on the formation of an electrically conductive network on the fabric surface after plating NPs. The electromagnetic interference (EMI) shielding efficiency (SE) increased to 15.5 dB after plating Cu NPs on the polyethylene terephthalate (PET) fabric. The ability to shield EM waves depended on forming an electrically conductive network on the fabric surface after plating Cu NPs [54]. The silver-nanoparticle-coated woven cotton fabric showed an EMI shielding effectiveness of −20 dB [55]. Compared to textiles coated with silver nanoparticles, the sample in this paper had the same excellent performance, making it a promising textile with many protective properties. 

## 3. Experiment

### 3.1. Materials

Anhydrous ethanol (AR grade) was obtained from Tianjin Hengxing Reagent Co., Ltd. (Tianjin, China). AR-grade PVB with a molecular weight of 80,000 was purchased from Shanghai Sinopharm Group Reagent Co., Ltd. (Shanghai, China). AR-grade AgNO_3_ was acquired from Tianjin Damao Chemical Reagent Factory (Tianjin, China). The high-voltage DC power source was purchased from Dongwen High-voltage Power Supply (Tianjin) Co., Ltd. (Tianjin, China). An LSP-10-18 microinjection pump was obtained from Baoding Lange Constant Flow Mercury Co., Ltd. (Baoding, China). An oil-cooled electric drum collecting device was purchased from Shandong Zibo Weirun Machinery Factory (Zibo, China).

### 3.2. Preparation

Solutions for electrospinning were prepared with different concentrations of AgNO_3_ (0%, 0.5%, 1%, 1.5%, and 2.0% wt%) and 7.375% wt% PVB. Each solution was magnetically stirred for 18 h to ensure the complete dissolution of AgNO_3_ and PVB, resulting in a transparent solution. The electrospinning process was conducted at an ambient temperature of approximately 25 °C and a relative humidity of 40–60%. The electrospinning parameters were set as follows: a collection distance of 12 cm; voltage of 22 kV; flow rate of 1.5 mL/h; and three 18 G needles, each with an inner diameter of 0.84 mm and arranged at 3 cm intervals. After 4 h of electrospinning, the electrospun AgNO_3_/PVB nanofiber membrane was collected. To induce the reduction of AgNO_3_ to Ag NPs with UV light, the AgNO_3_/PVB nanofiber membrane was irradiated with UV light for 60 min at a temperature of 60 °C and a power of 0.89 W/m^2^ using a QUV accelerated aging machine (Wenzhou Darong Textile Instrument Co., Ltd., Wenzhou, China) to prepare the Ag NP/PVB nanofiber membrane.

### 3.3. Characterization

The surface morphology of the nanofiber bundles was examined using a scanning electron microscope (SEM, JSM-6390, JEOL Co., Ltd., Tokyo, Japan). The XRD spectrum of the nanofiber was measured using an X-ray diffractometer (Rigaku D/max2550VB3, Rigaku Co., Ltd., Tokyo, Japan). The UV transmittance and anti-UV performance of the nanofiber films were assessed using a textile anti-UV performance tester (YG912E, Quanzhou Meibang Instrument Co., Ltd., Quanzhou, China). The anti-electromagnetic radiation performance of the films was evaluated using an anti-electromagnetic radiation performance tester (DR-913G, Wenzhou Darong Textile Instrument Co., Ltd.).

The preparation and characterization process of silver-nanoparticle PVB nanofibers are shown in Figure 1. The AgNO_3_/PVB nanofiber membrane was prepared via electrospinning first. Then, the AgNO_3_/PVB nanofiber membrane was exposed to UV light to form an Ag NP/PVB nanofiber film, and finally, tests and analyses were performed.

## 4. Conclusions

With the increasing frequency of extreme weather events worldwide, the demand for multifunctional textiles is rising. In this study, we prepared AgNO_3_/PVB nanofiber membranes, which was transformed into Ag NP/PVB nanofiber membranes under UV radiation. Through testing, it was found that both AgNO_3_/PVB and Ag NP/PVB had UV resistance and electromagnetic shielding properties, with their UV resistance being particularly exceptional. This makes them suitable for use in scenarios with high UV protection requirements. According to the literature, Ag NP/PVB nanofiber membranes also possess excellent antibacterial properties, making the samples prepared in this study multifunctional. 

Some interesting experimental phenomena were also observed during the research process. The addition of AgNO_3_ improved the conductivity of the spinning solution, resulting in a reduction in fiber diameter due to enhanced stretching caused by a larger electric field force. Under UV light, the fusion and adhesion of PVB fibers led to an increase in fiber diameter and irregularity. Nevertheless, the presence of AgNO_3_ in PVB fibers reduced the temperature increase rate of the nanofiber membranes by absorbing UV energy and undergoing a chemical reaction, thereby minimizing fiber fusion. Notably, a significant number of PVB fibers fused and adhered to each other upon reaching the glass transition temperature. The incorporation of AgNO_3_ effectively alleviated the fiber adhesion issue, thereby stabilizing the temperature increase in the irradiated PVB fibers. The XRD patterns exhibited characteristic diffraction peaks corresponding to Ag NPs, indicating their crystalline nature. The anti-UV performance of the nanofiber membranes was enhanced by the addition of AgNO_3_ or Ag NPs. The nanofiber film containing AgNO_3_-derived Ag NPs demonstrated better anti-UV performance compared to the non-irradiated nanofiber film containing AgNO_3_. The anti-electromagnetic radiation performance slightly differed between the nanofiber films containing AgNO_3_ and AgNO_3_-derived Ag NPs, with superior performance observed at the frequency of 2.5 GHZ for the nanofiber film containing AgNO_3_-derived Ag NPs. The nanofiber membrane displayed superior shielding effects against medium- and low-frequency electromagnetic waves compared to high-frequency waves.

The experimental phenomena and pattern analysis in the study process provided useful references for preparing multifunctional materials containing Ag NPs, especially by proposing methods to enhance the anti-UV function of existing materials. However, due to the limitations of the laboratory conditions, tests for the mechanical properties of the samples, as well as transmission electron microscopy (TEM) and X-ray photoelectron spectroscopy (XPS) tests, could not be completed. Consequently, the UV radiation impact on the mechanical properties of the samples had not been analyzed. Moreover, further analysis by TEM and XPS had not been conducted to determine the distribution and existence of silver nanoparticles. In the future, the project team will improve experimental conditions, prepare samples based on actual usage requirements, and conduct more comprehensive sample testing. The focus will be on optimizing the sample preparation methods to achieve the best mechanical performance. To this end, we will be using more comprehensive tests, such as TEM and XPS, to analyze the distribution of functional nanoparticles and the pattern of their impact on performance. This will expand its applications in other fields, such as catalysis, new electrode materials for batteries, electrically conductive materials, and low-temperature thermally conductive materials, to meet the highest multifunctional requirements. The team also has a concept: many new energy vehicles now have glass sunroofs, like Tesla, but the sun is too intense in the summer. It is worth further exploration whether silver nitrate can be added when using PVB as a laminated glass to enhance the UV resistance of the glass.

## Data Availability

The authors declare that the data supporting the findings of this study are available within the paper. Should any raw data files be needed in another format, they are available from the corresponding author [S. Cao] upon reasonable request.

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
