# Peer review of "Multi-Functional Electrospun AgNO3/PVB and Its Ag NP/PVB Nanofiber Membrane"

_molecules, 2023, doi:10.3390/molecules28166157_

Round 1

Reviewer 1 Report

Reviewer's  comments:

Before publication, the authors must correct the mistakes in the text that are found all over the manuscript and address the following:

1.  The authors should change the manuscript title, “Electrospun AgNO3/PVB and its Ag NPs/PVB Nanofiber Membrane” title is not appealing or attractive, and its suggestion of adding fabrication or characterization to the title.

2.     The authors should add targeted application in the manuscript.

3.   Why did the author use AgNO3/PVB while many researchers have already done that? Please, explain.

4.      Please explain the method (preparation of solution to the membrane development in detail with clarity), e.g., the AgNO3/PVB and Ag NPs/PVB membrane.

5.    The Introduction should consist of five paragraphs answering the following five questions: What is the problem? Why is it interesting and important? Why is it hard? Why hasn't it been solved before? (or, what's wrong with previously proposed solutions? What are the key components of my approach and results?

6.    It is not clear what is new in this study. The novelty of the presented work should be highlighted in a proper way. as other researchers have already conducted the same work and characterization of the same materials?

7.      It is better to insert numbers and references for each equation.

8.   Comparison between obtained results and literature data is a very weakness of this work. Besides, the obtained results are not compared with published data by other researchers. For more contribution, the Authors should compare their results with those in relevant published works of other researchers. If it is possible, The Authors should compare their results with those of previous works.

9.     What is the novelty in your work, please explain.

10.    Scheme 1 needs to be revised; it looks very complex. It might not be very pleasant to the reader.

11.    The author needs to measure the membrane's mechanical properties before and after incorporation of AgNo3 as well as before and after the influence of UV irradiation., also inform mainly which ASTM standard would be adopted.

12.  The author must perform TEM or XPS to confirm Ag NPs or their existence.

13.  Manuscript is too short, more information should be added, or result related to the manuscript work or application.

14.  The authors need to improve all figures' resultions.

15.  The author should add at least 3-5 new references from anywhere, especially from the "Molecules" relevant to his topics.

16.  Conclusions should be more concrete and future research directions presented

17.  Overall, the work is interesting; it just needs to follow the suggestions to improve the manuscript.

Reviewer's  comments:

Before publication, the authors must correct the mistakes in the text that are found all over the manuscript and address the following:

1.      The authors should change the manuscript title, “Electrospun AgNO3/PVB and its Ag NPs/PVB Nanofiber Membrane” title is not appealing or attractive, and its suggestion of adding fabrication or characterisation to the title.

2.      The authors should add targeted application in the manuscript.

3.      Why did the author use AgNO3/PVB while many researchers have already done that? Please, explain.

4.      Please explain the method (preparation of solution to the membrane development in detail with clarity), e.g., the AgNO3/PVB and Ag NPs/PVB membrane.

5.      The Introduction should consist of five paragraphs answering the following five questions: What is the problem? Why is it interesting and important? Why is it hard? Why hasn't it been solved before? (or, what's wrong with previously proposed solutions? What are the key components of my approach and results?

6.      It is not clear what is new in this study. The novelty of the presented work should be highlighted in a proper way. as other researchers have already conducted the same work and characterization of the same materials?

7.      It is better to insert number and reference for each equation.

8.      Comparison between obtained results and literature data is a very weakness of this work. Besides, the obtained results are not compared with published data by other researchers. For more contribution, the Authors should compare their results with those in relevant published works of other researchers. If it is possible, The Authors should compare their results with those of previous works.

9.      What is the novelty in your work, please explain?

10.  Scheme 1 needs to be revised; it looks very complex. It might not be very pleasant to the reader.

11.  The author needs to measure the membrane's mechanical properties before and after incorporation of AgNo3 as well as before and after the influence of UV irradiation., also inform mainly which ASTM standard would be adopted.

12.  The author must perform TEM or XPS to confirm Ag NPs or their existence.

13.  Manuscript is too short need to be add more information or result related to manuscript work or application.

14.  The author need to be improves all figures resultion.

15.  The author should need to add at least 3-5 new references from anywhere or specially the "Molecules" relevant to his topics.

16.  Conclusions should be more concrete and future research directions presented

17.  Overall, the work is interesting; it just needs to follow the suggestions to improve the manuscript.

Reviewer 2 Report

The manuscript entitled “Electrospun AgNO3/PVB and its Ag NPs/PVB Nanofiber membrane” shows interesting information for preparation of fibermembrane containing AgNO3 by using electrospinning technique. However, there are some comments and questions for this manuscript as follows:

1. The authors should specify the novelty of this study in the Introduction part of the manuscript.

2. What are the applications of the nanofilms prepared in this study? Please give some examples of applications of this fibers in the real situations (with or without exposure of UV) in the Introduction part.

3. The authors talked about PAN 2 times in the manuscript, could the authors give more details about this polymer in the Introduction part?

4. Although, this manuscript was well organized and gave a clear explanation for a mechanism of change of AgNO3 to be Ag nanoparticles, the authors should show the evidence of formation of Ag nanoparticles in the nanofibers.

5. Why are the authors sure that the different XRD spectra in Figure 4 were from the formation of Ag nanoparticles? The authors should add the results from investigation of the positive control for XRD study.

6. In particular, physicochemical properties of the obtained Ag nanoparticles, such as, particle size, polydispersity index, zeta potential and the morphology of Ag nanoparticles should also be included in this manuscript.

7. The authors should specify in the figure caption that which formulation provided the results shown in Figure 2(b).

8.  Please check a unit of frequency in Line 22, should it be 2.5 GHz?

Round 2

Reviewer 1 Report

Reviewer' 1 comments: 

Reviewer #: In the present work (Revised Manuscript Molecules-2539784-R2) "

"After a thorough review of the author's revised manuscript (Manuscript Molecules-2539784-R2), I conclude that they have provided his answers to the reviewers' suggested comments with appropriate explanations.

Therefore, I accept and recommend this version related to the Journal of Molecules." 

Reviewer 2 Report

The authors responded to the addressed comments and questions. This manuscript could thus be accepted for publication.